# Circulating Tumor DNA as a Real-Time Biomarker for Minimal Residual Disease and Recurrence Prediction in Stage II Colorectal Cancer: A Systematic Review and Meta-Analysis

**DOI:** 10.3390/ijms26062486

**Published:** 2025-03-11

**Authors:** Silvia Negro, Alessandra Pulvirenti, Chiara Trento, Stefano Indraccolo, Stefania Ferrari, Marco Scarpa, Emanuele Damiano Luca Urso, Francesca Bergamo, Salvatore Pucciarelli, Simona Deidda, Angelo Restivo, Sara Lonardi, Gaya Spolverato

**Affiliations:** 1Third Surgical Clinic, Department of Surgical, Oncological and Gastroenterological Sciences (DiSCOG), University of Padova, 35128 Padova, Italy; alessandra.pulvirenti@unipd.it (A.P.); stefania.ferrari10@gmail.com (S.F.); marco.scarpa@unipd.it (M.S.); edl.urso@unipd.it (E.D.L.U.); puc@unipd.it (S.P.); gaya.spolverato@unipd.it (G.S.); 2Department of Surgery, Oncology and Gastroenterology, University of Padova, 35128 Padova, Italy; chiara.trento@phd.unipd.it (C.T.); stefano.indraccolo@unipd.it (S.I.); 3Basic and Translational Oncology Unit, Veneto Institute of Oncology IOV-IRCCS, 35128 Padova, Italy; 4Department of Oncology, Veneto Institute of Oncology IOV-IRCCS, 35128 Padova, Italy; francesca.bergamo@iov.veneto.it (F.B.); sara.lonardi@iov.veneto.it (S.L.); 5Colorectal Surgery Unit, A.O.U. Cagliari, Department of Surgical Science, University of Cagliari, 09042 Cagliari, Italy; simonadeidda86@gmail.com (S.D.); arestivo@unica.it (A.R.)

**Keywords:** circulating tumor DNA, minimal residual disease, colorectal cancer, adjuvant chemotherapy, dynamic surveillance, precision oncology

## Abstract

The role of adjuvant chemotherapy (adj-CT) in stage II colon cancer remains controversial. Circulating tumor DNA (ctDNA) is a promising biomarker for detecting minimal residual disease (MRD) and predicting recurrence. This systematic review and meta-analysis evaluated the prognostic value of ctDNA in stage II colorectal cancer (CRC), focusing on postoperative detection, post adj-CT outcomes, and dynamic surveillance. A literature search identified studies correlating ctDNA positivity in stage II CRC with recurrence risk, recurrence-free survival (RFS), and disease-free survival (DFS). Seven studies met the inclusion criteria. Postoperative ctDNA positivity significantly increased the risk of recurrence (pooled risk ratio [RR:] 3.66; 95% confidence interval [CI]: 1.25–10.72; *p* = 0.002). CtDNA positivity after adj-CT was strongly associated with poor survival, while dynamic ctDNA monitoring detected recurrence earlier than conventional methods, including carcinoembryonic antigen (CEA) and imaging. CtDNA is a robust prognostic biomarker in stage II CRC, enabling personalized treatment. High-risk ctDNA-positive patients may benefit from intensified therapy, while ctDNA-negative patients could avoid unnecessary treatments. However, the standardization of detection methods and large-scale validation studies are needed before integrating ctDNA into routine clinical practice as a non-invasive, dynamic tool for personalized care.

## 1. Introduction

Colorectal cancer (CRC) is one of the most prevalent and deadly malignancies globally, accounting for 1.9 million new cases and 935,000 deaths annually [1]. Although surgery remains the primary curative treatment for resectable CRC, recurrence poses a significant challenge, affecting approximately 30–50% of patients. Notably, nearly 25% of recurrences occur in individuals initially diagnosed with stage II disease, which represents around 30% of all CRC cases [2,3]. The 8th edition of the TNM classification system stratifies stage II patients into low-risk and high-risk categories based on clinicopathological factors, such as lymphovascular invasion, poor tumor differentiation, and T4 tumor stage. High-risk patients are considered candidates for adjuvant chemotherapy (adj-CT) [4]. However, there is a significant lack of clarity regarding which patients will truly benefit from postoperative therapies. Many patients with high-risk features do not experience recurrence, while some considered low-risk unexpectedly relapse. This uncertainty leads to suboptimal treatment decisions, with a substantial proportion of patients unnecessarily exposed to the potential toxicities of adj-CT [5,6,7,8]. Therefore, these issues highlight the urgent need for a robust and more effective method for predicting treatment response and facilitate relapse detection. Minimal residual disease (MRD) is defined as the presence of a small number of neoplastic cells in the bloodstream after curative treatment. While these residual cells are not detectable using standard diagnostic techniques, they can be indirectly identified using highly sensitive methods, such as circulating tumor DNA (ctDNA) analysis [9]. ctDNA, which consists of short DNA fragments shed by apoptotic and necrotic tumor cells into the bloodstream, serves as a real-time molecular biomarker reflecting the presence of residual disease and tumor dynamics. Unlike traditional biomarkers, ctDNA allows for the non-invasive assessment of tumor burden, clonal evolution, and treatment response, making it an attractive tool for precision oncology. Recent advancements in sequencing technologies, such as digital droplet PCR (ddPCR) and next-generation sequencing (NGS), have significantly enhanced the sensitivity and specificity of ctDNA detection. By analyzing tumor-derived genetic material from peripheral blood, ctDNA can provide molecular evidence of recurrence months before its clinical or radiological confirmation. Several studies have demonstrated that ctDNA detection post-surgery is associated with a markedly higher risk of relapse, even in patients considered to be at low clinical risk. Moreover, ctDNA can capture the minimal tumor burden, which conventional imaging modalities fail to detect, reinforcing its role as a powerful prognostic indicator [10,11]. Observational studies in patients with solid tumors have shown a very high risk of recurrence when ctDNA is detected after curative-intent surgery [12,13,14]. However, its role in guiding treatment decisions in CRC remains controversial. Most existing studies have focused on metastatic disease and often fail to differentiate between CRC stages, particularly in the perioperative setting [15,16]. Emerging evidence suggests that ctDNA can be utilized for dynamic risk stratification, allowing for tailored treatment approaches based on real-time molecular assessment, rather than solely on clinicopathological features [17]. This individualized approach has the potential to refine adjuvant therapy recommendations, preventing overtreatment in patients who may not benefit from additional chemotherapy while ensuring that high-risk patients receive timely interventions. To address this knowledge gap, the present study conducts a systematic review and meta-analysis of recent research on the prognostic value of postoperative ctDNA in stage II CRC. Specifically, we will explore three main topics: (a) the association between postoperative ctDNA positivity and recurrence risk in patients who have undergone curative-intent surgery for CRC; (b) the prognostic value of ctDNA positivity following adj-CT; (c) the utility of ctDNA as a dynamic tool for monitoring recurrence during follow-up. This review will provide a comprehensive analysis of existing literature to clarify the clinical relevance of ctDNA as a biomarker for recurrence risk assessment and treatment stratification in stage II CRC. By addressing current knowledge gaps, this study aims to contribute to more precise and personalized therapeutic strategies for CRC management.

## 2. Materials and Methods

### 2.1. Study Registration

The study protocol was registered in the International Prospective Register of Systematic Reviews (PROSPERO) (registration number CRD42025632993) on 1 January 2025.

### 2.2. Search Strategy

A literature search was performed to identify studies reporting the association between ctDNA and stage II CRC, specifically evaluating the correlation between postoperative ctDNA positivity and recurrence risk in patients who underwent curative-intent surgery for CRC. A systematic review of EMBASE, MEDLINE, PubMed, and the Cochrane Central Register of Controlled Trials was conducted. The search strategy used the following medical subject headings (MeSH) and free-text keywords: (colorectal OR colon OR rectal) AND (cancer OR neoplasm OR adenocarcinoma) AND (ctDNA OR circulating tumor DNA OR cell-free DNA OR cf-DNA) AND (recurrence risk OR tumor relapse OR minimal residual disease OR MRD). No restrictions on publication date were applied. Two independent reviewers (S.N. and A.P.) conducted the literature review separately according to the established inclusion criteria. Data regarding study design, populations, setting, methods of ctDNA measurement (e.g., ddPCR, NGS), timing of ctDNA assessment (e.g., postoperative, preoperative), and reported outcomes (e.g., recurrence risk, recurrence-free survival (RFS), overall survival (OS)) were extracted and registered separately by the reviewers, and a database of the selected papers was created. After duplicates were removed, disagreements were resolved by two additional blinded reviewers (S.L. and G.S.). The systematic review was conducted in accordance with the Preferred Reporting Items for Systematic Reviews and Meta-Analyses (PRISMA) guidelines [18]. Given the nature of the study (a systematic review) and the absence of direct reporting of individual patient data, institutional review board approval was not required. The work was reported in accordance with the AMSTAR (assessing the methodological quality of systematic reviews) guidelines [19]. For detailed information, see the Appendix A.

### 2.3. Inclusion and Exclusion Criteria

Inclusion criteria were as follows: (a) prospective cohort studies, retrospective cohort studies, case-control studies, randomized controlled trials, systematic reviews, and meta-analyses with data that were relevant to ctDNA and stage II CRC recurrence; (b) studies evaluating the prognostic value of ctDNA in predicting recurrence in stage II CRC, and studies analyzing ctDNA as a marker of MRD after surgery; (c) human studies in adult populations with stage II CRC who underwent curative-intent surgery; (d) studies reporting disease-free survival (DFS), RFS, OS, or recurrence risk associated with ctDNA positivity. Exclusion criteria were as follows: (a) case reports, case series, editorials, reviews or meta-analyses without original data, preclinical studies, animal studies, or studies with in vitro analyses only; (b) studies focusing exclusively on metastatic CRC or stages I and III without clear data on stage II, studies not involving ctDNA or involving ctDNA in a way that is unrelated to its role in recurrence risk assessment; (c) studies including only patients with stage I, III, or IV CRC, or non-CRC, mixed-stage studies that do not separate outcomes for stage II CRC; (d) studies without clear outcomes to recurrence or survival, studies with insufficient or incomplete data on ctDNA positivity and recurrence risk; (e) non-English studies (unless a reliable translation was available), non-peer-reviewed literature (e.g., conference abstracts, theses, or gray literature).

### 2.4. Data Extraction and Assessment of the Quality of Included Studies

Based on the above inclusion and exclusion criteria, data were extracted using pro forma tables with prespecified variables. The dataset included the following variables: (a) study characteristics (author, years of publication, study design, country, and sample size); (b) population characteristics (e.g., sex); (c) intervention details (methods of ctDNA analysis and timing of measurement); and (d) outcomes (ctDNA positivity after surgery, ctDNA positivity following completion of adj-CT, and ctDNA positivity during follow-up). The quality of eligible studies was assessed using Newcastle–Ottawa (NOS) (for observational studies) and the Cochrane risk-of-bias tool (RoB 2) (for randomized controlled trials). Two independent reviewers assessed the risk of bias in each study. Any discrepancies were resolved through discussion with a third reviewer.

### 2.5. Statistical Methods

A meta-analysis was conducted to evaluate recurrence risk in postoperative ctDNA-positive stage II CRC patients using Review Manager 5.4 (Cochrane Collaboration, 2020). From each study, the number of ctDNA-positive and ctDNA-negative patients and the number of events (recurrences) in each category were extracted, and recurrence risk was calculated. In the cumulative meta-analyses, data were combined chronologically as each study was included. A combined risk ratio (RR) with 95% confidence intervals (CIs) was used to quantify this association. Heterogeneity across studies was assessed using the chi-squared Q test and I^2^ statistics. An I^2^ value less than 50% indicated low heterogeneity, in which case a fixed-effects model was applied. If heterogeneity exceeded this threshold (I^2^ ≥ 50%), a random-effects model was used to account for variability across studies. Sensitivity analyses were performed to evaluate the stability of the results by excluding individual studies (a leave-one-out approach) to assess their impact on the overall effect estimate. An assessment of publication bias and a funnel plot was not conducted, as our meta-analysis includes fewer than 10 studies.

## 3. Results

### 3.1. Study Characteristics and Quality

A total of 1097 studies were identified by the database search. After the removal of 6 duplicate records identified by Covidence, 1091 studies were identified by title and abstract. A total of 139 full-text articles were retrieved, of which 132 were excluded. Finally, seven studies met the eligibility criteria and were included in the systematic review [5,14,20,21,22,23,24]. Of these, six studies clearly reported data on ctDNA positivity and recurrence risk in stage II CRC and were considered suitable for inclusion in the meta-analysis [5,14,21,22,23]. The inter-rater agreement between reviewers was strong, with a Cohen’s kappa (κ) of 0.79. The diagram of the search results is shown in Figure 1. Of the included studies, five were multi-center and two were single-center studies, five were prospective observational studies, one was retrospective, and one was a randomized controlled trial. Table 1 summarized the main characteristics of the included studies. All studies evaluated recurrence risk in CRC cancer patients, except for the two studies by Tie et al. [5,14], which focused exclusively on colon cancer patients. In all seven included studies, researchers analyzed ctDNA in patients’ plasma after surgery. Additionally, five studies examined ctDNA’s role following adj-CT, and four included serial ctDNA measurements during follow-up. Of the seven studies, all used a tumor-informed method for ctDNA detection, except for Yang et al. [21], which also utilized a tumor-agnostic approach, and Nakamura et al. [22], which exclusively used a tumor-agnostic approach. In some cases, the recurrence risk specific to stage II CRC was calculated from supplementary data provided by the studies. Among the included studies, there is some variability in the definition of a “ctDNA-positive” or “ctDNA-negative” sample, as the authors used different methods and thresholds. The quality of the included studies was evaluated using the Newcastle—Ottawa Scale (NOS) for the six observational studies and the risk-of-bias 2 (RoB 2) tool scale for the single randomized controlled trial. The quality assessment results are summarized in Table 2. All seven studies scored at least eight stars on the NOS and were rated as having a low risk of bias according to the RoB 2, which we considered indicative of high methodological quality.

### 3.2. Recurrence Risk According to Postoperative ctDNA Status

Out of seven studies reporting comparative recurrence risk analysis, six were included in the meta-analysis [5,14,21,22,23,24]. Schøler et al. was excluded due to its limited evidence for stage II CRC, attributed to a small sample size [20,21]. Thus, a meta-analysis comparing risk of recurrence for postoperative ctDNA-positive and ctDNA-negative patients was conducted. Overall, the meta-analysis included a total of 126 ctDNA-positive patients and 964 ctDNA-negative patients. The pooled analysis, using a random-effects model, revealed that postoperative ctDNA-positive patients had a significantly higher risk of recurrence compared to ctDNA-negative patients, with a pooled risk ratio (RR) of 3.66 (95% CI: 1.25–10.72; *p* = 0.002). All the included studies individually showed an increased risk of recurrence for ctDNA-positive patients, with the exception of Yang et al. [21], which reported a non-significant risk ratio of 0.70 (95% CI: 0.21–2.31), contributing 13.8% of the total weight. High heterogeneity was observed among the included studies (Tau^2^ = 0.85; Chi^2^ = 48.52, df = 5; *p* < 0.00001; I^2^ = 90%) Figure 2.

### 3.3. Subgroup Analysis

Due to high heterogeneity observed between the six included studies, we conducted a subgroup analysis of patients stratified by potential confounding factors: the ctDNA measurement method (tumor-informed vs. tumor-agnostic), tumor site (colon vs. rectum) and chemotherapy status (adj-CT vs. CT-naive).

#### 3.3.1. Subgroup Analysis by the ctDNA Measurement Method

The forest plot in Figure 3 presents the pooled effects for recurrence risk in stage II CRC patients, focusing exclusively on studies that utilized a tumor-informed ctDNA measurement approach [5,14,21,23,24]. For Yang et al., only data from tumor-informed ctDNA measurements were extracted [21]. A total of 102 ctDNA-positive patients and 887 ctDNA-negative patients were included in this analysis. The pooled RR for recurrence among ctDNA-positive patients measured using a tumor-informed method was 4.87 (95% CI: 2.22–10.66; *p* < 0.0001), although high heterogeneity was found (Tau^2^ = 0.70; Chi^2^ = 41.34, df = 4; *p* < 0.00001; I^2^ = 90%).

#### 3.3.2. Subgroup Analysis by Tumor Site

Figure 4 presents the forest plot of pooled effects for recurrence risk in stage II CRC patients, focusing exclusively on studies evaluating patients with colon cancer [5,14,21,22]. A total of 73 ctDNA-positive patients and 542 ctDNA-negative patients were included. The pooled analysis demonstrated a significantly higher risk of recurrence for ctDNA-positive patients compared to ctDNA-negative patients, with a pooled RR of 4.93 (95% CI: 3.34–7.28; *p* < 0.00001). Low heterogeneity was observed in this subgroup analysis, with I^2^ = 35% (Chi^2^ = 4.59, df = 3; *p* = 0.20).

#### 3.3.3. Subgroup Analysis by adj-CT status

Figure 5 presents the forest plot of pooled effects for recurrence risk in stage II CRC patients, focusing exclusively on those who were CT-naive [14,22,24]. A total of 18 ctDNA-positive patients and 239 ctDNA-negative patients were included in this subgroup analysis. The pooled analysis revealed a significantly higher recurrence risk for ctDNA-positive patients compared to ctDNA-negative patients, with a pooled RR of 5.58 (95% CI: 3.43–9.10; *p* < 0.00001). Low heterogeneity was observed, with I^2^ = 26% (Chi^2^ = 2.71, df = 2.71; *p* = 0.26).

### 3.4. Recurrence Risk According to Post adj-CT ctDNA Status

Five of the seven studies included in the systematic review evaluated the role of ctDNA after the completion of adjuvant chemotherapy. Tie et al. reported that patients who were ctDNA positive after completing CT had significantly worse RFS compared to ctDNA-negative patients, with a hazard ratio (HR) of 11.0 (95% CI: 1.8–68; *p* = 0.001) [14]. Similarly, Kotani et al. found that ctDNA positivity after the completion of chemotherapy was strongly associated with poor DFS across all stages, with an HR of 11.0 (95% CI: 5.2–23; *p* < 0.0001) [23]. Nakamura et al. confirmed these findings, demonstrating a strong correlation between ctDNA positivity after adjuvant chemotherapy and poor RFS, with an HR of 11.58 (95% CI: 1.33–101; *p* = 0.001) [22]. In addition, Schøler et al. found that ctDNA levels decreased in correlation with tumor volume reduction during adj-CT, highlighting the dynamic nature of ctDNA as a biomarker across all stages [20]. Limited evidence was found in Yang et al.’s study due to its small sample size [21].

### 3.5. Recurrence Risk According to ctDNA Status During Follow-Up

Four of the seven studies included in the systematic review evaluated the role of serial ctDNA measurements during follow-up. Tie et al. reported that ctDNA positivity during follow-up was significantly superior to CEA in detecting recurrence, with a sensitivity of 85% compared to 41% (*p* = 0.002). Furthermore, ctDNA positivity predicted recurrence earlier than CEA, with a median lead time of 61 days compared to 167 days (*p* = 0.04) [14]. Schøler et al. highlighted the advantage of ctDNA over radiological imaging for recurrence detection, showing a significantly shorter time between ctDNA detection and recurrence, with a median of 8.2 months compared to 16.9 months (*p* = 0.001) across all stages [20]. Kotani et al. found that patients who converted from a ctDNA-negative to ctDNA-positive status during follow-up exhibited a markedly higher risk of recurrence, with an HR of 14.0 (95% CI: 8.5–24.0; *p* < 0.001) [23]. Similarly, Grancher et al. observed that the median time between ctDNA detection and clinical recurrence was 12.8 months [24].

## 4. Discussion

This study demonstrates that ctDNA analysis in peripheral blood samples holds significant promise as a prognostic, predictive, and monitoring tool for CRC. By facilitating the early identification of patients with MRD, ctDNA testing could help inform decisions regarding systemic therapy, sparing ctDNA-negative patients from unnecessary treatments and their associated toxicities [25]. To our knowledge, this systematic review is the first to specifically evaluate the role of ctDNA in stage II CRC. The studies included consistently show that postoperative detection of ctDNA is strongly correlated with poor oncologic outcomes and, accordingly, reflects residual tumor burden and recurrence in stage II CRC. In contrast, preoperative ctDNA was not included in this review, as its prognostic value remains less well defined. While preoperative ctDNA levels may reflect the tumor burden, they do not necessarily predict postoperative MRD or recurrence risk after curative-intent surgery. Furthermore, preoperative ctDNA positivity does not always distinguish between localized and micrometastatic disease, making it less clinically actionable for recurrence prediction. Therefore, this review specifically focused on postoperative ctDNA, which more directly informs prognosis, recurrence risk, and treatment decisions in stage II CRC [26,27]. Currently, multiple ctDNA-based randomized clinical trials are underway to establish its clinical utility in stage II CRC, such as the COBRA trial (Van Morris, presented at ASCO), PRODIGE 70-Circulate, MEDOCC-CrEATE, and Circulate AIO-KRK-0217 [28,29,30,31], in addition to some prospective interventional trials enrolling high-risk stage II along with stage III patients (PEGASUS Trial) [32]. Promising findings have also emerged from updated data from the CIRCULATE-Japan GALAXY observational study [33]. This analysis, which included 2240 patients with stage II-III colon cancer and stage IV CRC, strongly reinforced the prognostic value of ctDNA positivity with significantly inferior DFS (HR: 11.99, *p* < 0.0001) and OS (HR: 9.68, *p* < 0.0001). Finally, the heterogeneity observed in our pooled analysis likely reflects variability in study design, ctDNA detection methods, and patient populations. Further research with standardized methods is needed.

### 4.1. Impact of ctDNA Measurement Techniques

CtDNA can be monitored using two primary approaches: tumor-informed and tumor-agnostic methods. Tumor-informed assays identify patient-specific genetic alterations, which are subsequently analyzed in plasma samples, whereas tumor-agnostic assays detect ctDNA without prior knowledge of the tumor’s molecular profile [34]. Historically, tumor-agnostic assays were considered less sensitive [35]. In our review, Yang et al. utilized both tumor-informed and tumor-agnostic approaches [36]. Notably, using the tumor-agnostic method, they failed to find a correlation between ctDNA positivity and recurrence. However, when focusing only on patients evaluated with the tumor-informed approach, the pooled risk ratio for ctDNA-positive patients was 4.67 (95% CI: 2.22–9.84; *p* < 0.0001). These findings align with those of Martínez-Castedo et al., who reported superior sensitivity and specificity in tumor-informed methods for predicting CRC recurrence, attributed to their personalized tumor profile designs. In contrast, Nakamura et al. used a tumor-agnostic method based on the tissue-free epigenomic MRD assay (Guardant Reveal), which utilized advanced epigenomic analysis focusing on over 20,000 tumor-specific hypermethylated regions [22]. In this study, the Guardant Reveal achieved a sensitivity of 81% in patients with stage II or higher, and a specificity exceeding 98% for residual disease detection. However, these promising results were challenged in the COBRA trial, which was stopped early due to negative findings regarding the efficacy of adjuvant chemotherapy versus surveillance in patients with ctDNA-positive stage IIA colon cancer following surgery. In the COBRA trial, the Guardant Reveal assay demonstrated a sensitivity of 56% and specificity of 95%, with a significant risk of false-positive results that may have biased the results [31]. These findings underscore the biological complexity of ctDNA and the technical limitations of current diagnostic assays, highlighting the need for further validation and technological advancements [37]. In our systematic review, tumor-informed ctDNA detection methods were widely used to track minimal residual disease (MRD) and assess recurrence risk in stage II CRC. These approaches rely on patient-specific genomic alterations identified through tumor tissue sequencing, which are subsequently monitored in plasma samples. The most frequently reported mutations included TP53, KRAS, NRAS, APC, and PIK3CA, reflecting their well-established role in CRC pathogenesis. Additionally, some studies incorporated BRAF V600E, a variant associated with aggressive tumor behavior and poor prognosis. The methodologies used to detect these variants varied, with NGS and ddPCR being the most common. NGS-based tumor-informed assays provide high sensitivity and allow for the detection of low-frequency variants, while ddPCR offers a highly specific and cost-effective alternative for tracking predefined hotspot mutations. However, differences in mutation panels, sequencing depth, and detection thresholds across studies introduce variability in ctDNA sensitivity and specificity. This highlights the need for standardization of tumor-informed ctDNA assays to ensure reproducibility and facilitate widespread clinical adoption. Future research should focus on refining these methods and exploring the integration of additional molecular markers, such as epigenetic modifications and fragmentomic analyses, to enhance the predictive power of ctDNA for MRD detection and recurrence monitoring.

### 4.2. Role of ctDNA by Tumor Site

All studies included in our systematic review analyzed patients without distinguishing between colon and rectal cancer, except for two studies by Tie et al. [5,14]. An analysis based on these two studies, plus supplementary data from Nakamura et al. and Yang et al. [21,22], revealed a pooled RR of 4.93 (95% CI: 3.34–7.28; *p* < 0.00001) with low heterogeneity (I^2^ = 35%). Stratification by tumor site is particularly important given the distinct biological and clinical differences between colon and rectal cancer. To date, no studies have directly compared the use of ctDNA between colon and rectal cancer. In addition, rectal cancer might have received CT with other treatments before surgery, and this is an important confounding factor. However, a previous study by Tie et al. on stage III colon cancer reported similar findings. In this study, ctDNA was detectable in 20 of 96 (21%) postsurgical samples and was significantly associated with inferior recurrence-free survival (HR: 3.8; 95% CI: 2.4–21.0; *p* < 0.001) [38].

### 4.3. Role of ctDNA in Guiding adj-CT

Numerous studies have demonstrated that MRD-positive patients, identified by detectable ctDNA, almost universally experience recurrence if not treated with systemic therapy. Detectable ctDNA is not only a high-risk marker but also a direct indicator of persistent disease. Actually, 95–100% of patients with persistently detectable ctDNA after surgery will develop a recurrence, typically within two years of follow-up, if systemic therapy is not administered [38,39]. Our results align with these observations, showing a pooled RR of 5.58 (95% CI: 3.43–9.10; *p* < 0.00001) in stage II ctDNA-positive, CT-naive patients. Similarly, the previously reported GALAXY study on stage II–III CRC identified postoperative ctDNA positivity as the most significant prognostic factor associated with poor OS, outperforming other well-established clinicopathological features [33]. Currently, several trials are evaluating the efficacy of ctDNA-guided adj-CT in early-stage CRC. The VEGA phase III study is assessing the non-inferiority of observation alone compared to standard adj-CT in patients with high-risk stage II or low-risk pathological stage III CRC confirmed to be ctDNA-negative four weeks after surgery [40]. Other multi-center phase II trials include the PEGASUS trial (focused on stage II-T4N0/III colon cancer), the NRG-GI005 (COBRA) trial (investigating low-risk stage II colon cancer) and the DYNAMIC trial (targeting stage II colon cancer) [31,32,41]. These trials are designed to validate ctDNA-guided approaches as an alternative to standard treatments, potentially refining treatment strategies and improving outcomes in early-stage CRC. Moreover, the presence of detectable ctDNA at the completion of adj-CT is strongly associated with poor survival and a high risk of disease recurrence. In this systematic review, ctDNA positivity after CT in early-stage CRC consistently emerged as a robust predictor of adverse outcomes. Henriksen et al. reported similar significantly shorter recurrence-free survival (HR: 94.2, *p* < 0.001) in stage III CRC patients who underwent adj-CT but failed to achieve ctDNA clearance [42]. Comparably, the IDEA-France study demonstrated improved DFS in stage III CRC patients with detectable ctDNA who received six months of adj-CT compared to those treated for shorter durations [43]. The consistent association between ctDNA positivity and adverse outcomes underscores its role as a reliable biomarker of residual disease and risk of recurrence following CT. Incorporating ctDNA analysis into post-treatment surveillance protocols could provide a valuable tool for identifying patients who may benefit from closer monitoring or intensified therapeutic interventions.

### 4.4. Dynamic Monitoring of ctDNA

CtDNA has emerged as a promising noninvasive biomarker capable of detecting MRD earlier than traditional methods, such as CT scans and carcinoembryonic antigen (CEA) testing [38,44,45]. In our systematic review, Tie et al. demonstrated that ctDNA has significantly superior sensitivity (85%) compared to CEA (41%) for detecting recurrence in stage II colon cancer. Furthermore, ctDNA can predict recurrence a median of 61 days earlier than CEA [14]. Similarly, Schøler et al. reported that ctDNA identified recurrence approximately eight months earlier than radiographic imaging (*p* = 0.001) [20]. Integrating ctDNA analysis with cross-sectional imaging could enhance post-treatment surveillance, offering a more dynamic and sensitive approach to monitoring prognosis. Prospective studies are needed to validate these findings and optimize surveillance protocols.

### 4.5. Limitations and Future Directions

A key limitation of postoperative ctDNA testing is the relatively low proportion of patients with detectable ctDNA after surgery, which may impact its overall clinical utility. While multiple studies have demonstrated that ctDNA positivity is a strong predictor of recurrence, the majority of patients in stage II CRC cohorts are ctDNA-negative postoperatively, yet some of these individuals will still experience disease relapse. This suggests that ctDNA negativity does not equate to the complete absence of residual disease and that other biological and clinicopathological factors may contribute to recurrence risk. Potential explanations for false-negative results include low ctDNA shedding by certain tumors, limited assay sensitivity, and the timing of blood collection post-surgery. Additionally, some patients may harbor micrometastatic diseases that are not captured by current ctDNA detection thresholds. Given these limitations, ctDNA should be interpreted in conjunction with other prognostic markers, such as tumor histopathology, molecular subtyping, and immune signatures, to refine risk stratification and optimize treatment decisions. Future studies should focus on enhancing ctDNA assay sensitivity and exploring integrated biomarker models that combine ctDNA with additional risk predictors to improve recurrence detection and guide personalized therapy in stage II CRC.

While this review underscores ctDNA’s potential, several challenges remain. Heterogeneity in ctDNA detection methods, study designs, and patient populations limits the generalizability of findings. Standardizing ctDNA assays, particularly tumor-informed versus tumor-agnostic methods, and harmonizing detection thresholds across studies are critical for clinical reproducibility. Additionally, the cost and accessibility of ctDNA testing remain barriers to widespread adoption. Future research should prioritize large-scale randomized controlled trials focusing on stage II CRC, with attention to patient demographics, tumor site-specific analysis, and long-term outcomes. Further exploration of cost-effectiveness and integration into clinical workflows is essential to facilitate routine use.

## 5. Conclusions

This study highlights the potential of ctDNA to address the limitations of current risk stratification systems by providing a personalized approach to the management of stage II CRC. Tumor-informed and advanced tumor-agnostic methods, site-specific testing, and integration of ctDNA into both pre- and post-treatment decision-making hold particular promise. However, standardization and validation in larger, prospective studies are necessary to enable its widespread clinical adoption. Addressing these challenges will ensure ctDNA’s role as a cornerstone of personalized care in early-stage CRC.

## Figures and Tables

**Figure 1 ijms-26-02486-f001:**
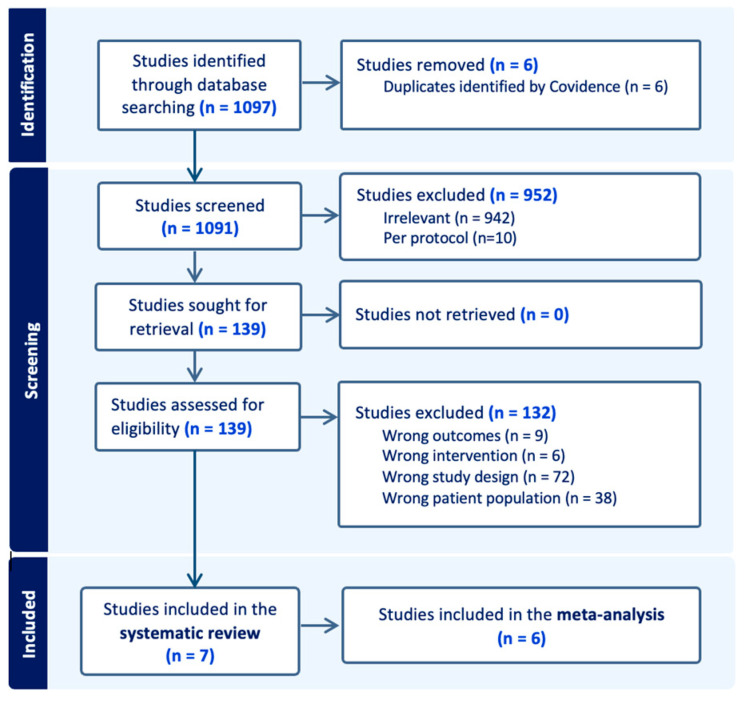
PRISMA flowchart of the included studies.

**Figure 2 ijms-26-02486-f002:**
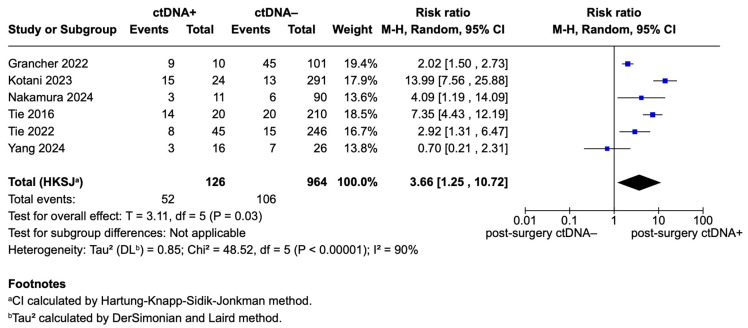
Forest plot of pooled effects for recurrence risk for ctDNA+ and ctDNA– stage II CRC patients [5,14,21,22,23,24].

**Figure 3 ijms-26-02486-f003:**
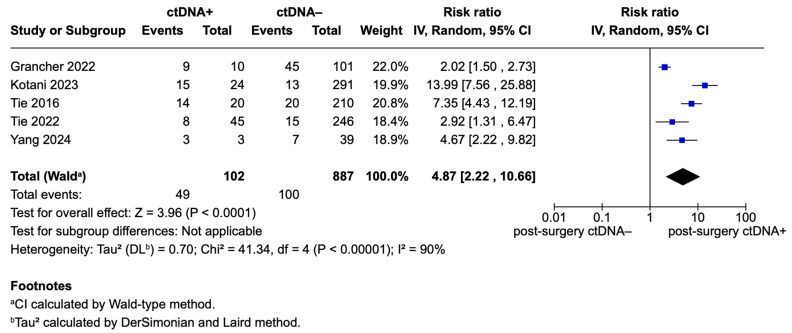
Forest plot of pooled effects for recurrence risk for ctDNA+ and ctDNA– stage II CRC patients stratified for ctDNA measurement method (only tumor-informed) [5,14,21,23,24].

**Figure 4 ijms-26-02486-f004:**
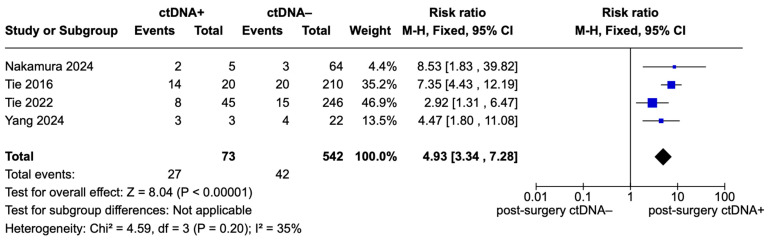
Forest plot of pooled effects for recurrence risk for ctDNA+ and ctDNA– stage II CRC patients stratified for tumor site (only colon) [5,14,21,22].

**Figure 5 ijms-26-02486-f005:**
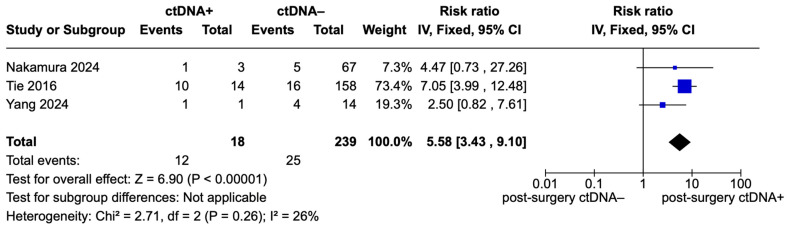
Forest plot of pooled effects for recurrence risk for ctDNA+ and ctDNA– stage II CRC patients stratified for ctDNA chemotherapy status (only CT-naive) [14,21,22].

**Table 1 ijms-26-02486-t001:** Main characteristics of the included studies.

Study, Year (Country)	Design	Stage (n, %), Site	No. Patients (Male/Female)	Sample Origin	Tumor-Informed vs. Tumor-Agnostic/Naive	Methods of ctDNA Measurement	ctDNA Sampling Time	ctDNA-Positive Rate (n/N)	Recurrence Risk (Metric)	Follow-Up Duration	Endpoints	ctDNA+ Post- Operative	ctDNA+ After CT	ctDNA+ During f.up
**Tie et al., 2016 (Australia)** [14]	Multi-center, prospective observational	Stage II: 230 (100%), colon	230 (131/99)	Plasma	Tumor-Informed	Droplet Digital PCR (ddPCR)	Post-op (4–10 weeks after surgery)	Post-op (overall): 20/230 (9%) Post-op (CT-naive): 14/178 (8%)	RR (overall): 7.00 (95% CI: 4.09–11.37; *p* < 0.0001) RR (CT-naive): 7.32 (95% CI: 3.93–12.60; *p* < 0.0001)	Median: 27 months (range 2–52)	RFS in ctDNA+ patients	ctDNA+ postoperative patients had a significantly higher recurrence risk.	ctDNA+ after completion of CT is associated with poor RFS (HR, 11; 95% CI: 1.8–68; *p* = 0.001)	ctDNA-+ during f.up is superior to CEA to detect the recurrence (85% vs. 41%; *p* = 0.002) Shorter time between ctDNA detection and recurrence in comparison to CEA (61 days vs. 167 days; *p* = 0.04)
**Schøler et al., 2017 (Denmark)** [20]	Single-center, prospective observational	Stage II: 9 (33%), CRC	27 (27/0)	Plasma	Tumor-Informed	Droplet Digital PCR (ddPCR) and Next-Gen Sequencing (NGS)	Post-op (day 8, day 30) and every 3 months up to 36 months	Pre-op: 4/8 (50%) Post-op: NA	NA	Median: 16.7 months (range 0.49–24.8)	RFS	Limited evidence for stage II CRC due to small sample size.	ctDNA+ level reduced according to tumor volume reduction during adj-CT (all stages).	Shorter time between ctDNA detection and recurrence in comparison to radiological imaging (8.2 months vs. 16.9 months; *p* = 0.001) (all stages).
**Tie et al., 2022 (Australia)** [5]	Multi-center, randomized controlled trial (evaluating ctDNA in guiding adj-CT in stage II colon cancer patients)	Stage II: 441 (100%): 294 (66%) (ctDNA-guided group), 147 (34%) (standard of care group), colon	441 (206/235)	Plasma	Tumor-Informed	Next- Generation Sequencing (NGS)	Post-op (4–7 weeks after surgery)	Post-op *: 45/291 (15%) * all ctDNA+ patients received adj-CT	RR (ctDNA + group): 2.91 (95% CI: 1.31–6.21; *p* < 0.01)	Median: 37 months	Assess the efficacy of a ctDNA-guided treatment strategy	ctDNA-guided treatment reduced adj-CT use without compromising RFS.	NA	NA
**Grancher et al., 2022 (France)** [24]	Multi-center, retrospective matched case-control (recurrent vs. non-recurrent)	Stage II (111, 100%), CRC (67 patients per arm)	111 (69/42)	Plasma	Tumor-Informed	Droplet Digital PCR (ddPCR)	Post-op (median: 81 and 99 days after surgery in the recurrent and non-recurrent groups)	Recurrent: 9/54 (17%) Non-Recurrent: 1/57 (2%)	OR = 11.13 (95% CI: 1.33–92.91; *p* = 0.03)	Median: 6.5 years	RFS	Postoperative ctDNA positivity was significantly associated with a higher recurrence risk. Median time from ctDNA detection to recurrence: 12.8 months.	NA	Median time between ctDNA detection and recurrence was 12.8 months.
**Kotani et al., 2023 (Japan)** [23]	Multi-center, prospective observational	Stage II (291, 34%), CRC	852 (433/491)	Plasma	Tumor-Informed	Next-Generation Sequencing (NGS)	Post-op (4 weeks after surgery)	Post-op: 24/315 (8%)	RR = 13.99 (95% CI: 7.46–25.40, *p* < 0.0001)	Median: 16.7 months (range 0.49–24.8)	RFS	ctDNA-positive patients had significantly higher recurrence rates and shorter RFS compared to ctDNA-negative patients.	ctDNA+ after completion of CT is associated with poor DFS (HR, 11; 95% CI: 5.2–23; *p* < 0.0001) (all stages).	A higher risk of recurrence was observed for patients who converted from ctDNA-negative to -positive during f.up (HR 14.0, 95% CI: 8.5–24.0, *p* < 0.001 (all stages).
**Yang et al., 2024 (China)** [21]	Single-center, prospective observational	Stage II: 42 (56%), CRC	42 (21/21)	Plasma	Tumor-Informed and Tumor-Agnostic	Next- Generation Sequencing (NGS)	Post-op (day 7 after surgery)	Stage II: Tumor-informed: ctDNA+: 3/42 ctDNA−: 39/42 Tumor-agnostic: ctDNA+: 16/42 ctDNA−: 26/42	Recurrence risk (tumor-informed): ctDNA+ rec: 3/3 (100%) ctDNA– rec: 7/39 (18%) RR = 5.57 (95% CI 2.06–10.72, *p* = 0.01) Recurrence risk (tumor-agnostic): ctDNA+ rec: 3/16 (19%) ctDNA– rec: 7/26 (27%) RR = 0.69 (95% CI 0.21–2.07, *p* = 0.71)	Median: 34 months (range 16–45)	Evaluate fixed-panel feasibility for MRD in CRC	Confirmed the prognostic value of post-op ctDNA positivity (tumor-informed panel) as an MRD indicator.	Limited evidence for CT in ctDNA+ due to small sample size.	NA
**Nakamura et al., 2024 (Japan)** [22]	Multi-center, prospective non-randomized	Stage II: 101 (29%), CRC	342 (190/152)	Plasma	Tumor-Agnostic	Tissue-free epigenomic MRD assay	Post-op (day 28) and every 3 to 6 months for 5 years	Post-op: 13/99 (13%)	RR = 5.6 (95% CI: 1.66–16.02; *p* = 0.003)	Median: 28.4 months (range 8.9–38)	RFS	Post-op ctDNA + strongly correlated with recurrence.	ctDNA+ after completion of CT is associated with poor RFS (HR, 11.58; 95% CI: 1.33–101; *p* = 0.001) (all stages).	NA

**Abbreviations:** adj-CT: adjuvant chemotherapy; CEA: carcinoembryonic antigen; CI: confidence interval; CRC: colorectal cancer; CT: chemotherapy; ctDNA: circulating tumor DNA; DFS: disease-free survival; f.up: follow-up; HR: hazard ratio; MRD: minimal residual disease; NA: not available; NGS: next-generation sequencing; NA: not applicable; OR: odds ratio; PCR: polymerase chain reaction; Post-op: postoperative; RFS: recurrence-free survival; RR: risk ratio.

**Table 2 ijms-26-02486-t002:** Quality assessment of the included studies.

**Results of the Newcastle**–**Ottawa Scale (NOS) Quality Assessment**							
**Authors**	**Year**	**Selection**		**Comparability**		**Outcome**	**Total**
Tie et al. [14]	2016	****		*		***	*********
Schøler et al. [20]	2017	****		*		***	*********
Grancher et al. [24]	2022	***		*		***	********
Kotani et al. [23]	2023	***		*		***	********
Nakamura et al. [22]	2024	****		*		***	*********
Yang et al. [21]	2024	***		*		***	********
**Results of the Risk of Bias 2 (ROB2) Quality Assessment**							
**Authors**	**Year**	**D1**	**D2**	**D3**	**D4**	**D5**	**Overall**
Tie et al. [5]	2022	Low	Low	Low	Low	Low	Low

D1: randomization process; D2: deviations from the intended interventions; D3: missing outcome data; D4: measurement of the outcome; D5: selection of the reported results. NOS legend: Total 9 stars (*********): High-quality study (low risk of bias); 8 stars (********): Moderate-quality study (some risk of bias); *, ***, ****: Low-quality study (higher risk of bias).

## Data Availability

No new data were created or analyzed in this study. Data sharing is not applicable to this article.

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
