# Peer review of "Circulating Tumor DNA as a Real-Time Biomarker for Minimal Residual Disease and Recurrence Prediction in Stage II Colorectal Cancer: A Systematic Review and Meta-Analysis"

_ijms, 2025, doi:10.3390/ijms26062486_

Round 1

Reviewer 1 Report

Comments and Suggestions for Authors

In this review, the prognostic utility of postoperative measurements of cell-free tumor DNA in stage II colorectal cancer patients is evaluated. Seven original studies are included, and the main finding is that detectable cell-free DNA after treatment is associated to worse prognosis.

The manuscript is well-written, and the text is easy to follow. I have some concerns and suggestions, listed below.

  • The aim/main topics are: “a) the association between postoperative ctDNA positivity and recurrence risk in patients who have undergone curative-intent surgery for CRC; b) the prognostic value of ctDNA positivity following adj-CT; c) the utility of ctDNA as a dynamic tool for monitor recurrence during follow- up.”
    I am not sure how well the topic (c) is covered by the review. MRD, which is described in the Introduction, is not really penetrated except for section 3.5, which does not focus on sensitivity and MRD measurements, but rather recurrence detection in general.
  • Why is only the post-operative state investigated? From a clinical point of view, the pre-operative sample would be more convenient. In the post-operative sample, the non-tumor cell-free DNA could potentially increase, “diluting” the ctDNA and decreasing the detection rate.
  • Carefully check all aberrations are spelled out only the first time they are used (for instance CEA on line 381), and listed below each figure and table (for instance “f.up” in Table 1).
  • I find the layout of Table 1 hard to follow. Should it be two tables, or can the information be fused into one table?
  • Are there other studies, for other carcinomas, that this study could be compared to? Are there any studies suggesting the post-operative sample might not be associated to prognosis? Any pre-operative investigations from stage II CRC?
  • The number of patients with positive post-operative ctDNA are few. The usefulness of this test thereby decreases, since most patients will not have a positive sample, and within that “negative” group there will still be patients with a high risk for relapse. This is not covered in the Discussion.
  • In the tumor-informed approaches, which targets did they find? Any recurrent hotspot variants? Recurrent genes?
  • Minor comment: Line 126-127, please check tense.

Author Response

Dear Reviewer 1,

Thank you very much for your thoughtful review and for your positive comments on our manuscript. We truly appreciate your time and effort in evaluating our work.

We have carefully considered your concerns and suggestions and have addressed each point accordingly. We believe that the revisions we have made have strengthened the manuscript, and we hope they meet your expectations.

Please do not hesitate to let us know if any further clarifications are needed.

Reviewer 1): Comments and Suggestions for Authors

In this review, the prognostic utility of postoperative measurements of cell-free tumor DNA in stage II colorectal cancer patients is evaluated. Seven original studies are included, and the main finding is that detectable cell-free DNA after treatment is associated to worse prognosis.

The manuscript is well-written, and the text is easy to follow. I have some concerns and suggestions, listed below.

Red (Reviewer) Green (Author)

  • The aim/main topics are: “a) the association between postoperative ctDNA positivity and recurrence risk in patients who have undergone curative-intent surgery for CRC; b) the prognostic value of ctDNA positivity following adj-CT; c) the utility of ctDNA as a dynamic tool for monitor recurrence during follow- up.” 
    I am not sure how well the topic (c) is covered by the review. MRD, which is described in the Introduction, is not really penetrated except for section 3.5, which does not focus on sensitivity and MRD measurements, but rather recurrence detection in general. In section 3.5 (Recurrence Risk according to ctDNA status during follow-up), we summarize findings from four studies that assessed the role of serial ctDNA measurements for recurrence detection. These studies demonstrated that ctDNA positivity during follow-up correlates strongly with disease recurrence and provides earlier detection compared to conventional surveillance tools such as carcinoembryonic antigen (CEA) and radiological imaging. We highlight that ctDNA can anticipate recurrence by several months, reinforcing its potential utility for real-time disease monitoring. Regarding MRD assessment, we discuss its significance in the Introduction as a key concept in ctDNA-based recurrence prediction. While section 3.5 primarily focuses on recurrence detection, MRD detection inherently underlies this process, as ctDNA positivity during follow-up reflects persistent disease at a molecular level before clinical relapse is evident. Furthermore, we emphasize that some studies have explored MRD detection sensitivity by comparing tumor-informed versus tumor-agnostic approaches, with tumor-informed assays showing superior predictive performance. We acknowledge that a more in-depth discussion of ctDNA sensitivity for MRD detection could further enhance the review. However, rather than focusing on MRD sensitivity as an isolated metric, our manuscript integrates this concept within the broader discussion of ctDNA dynamics, recurrence risk stratification, and treatment monitoring.
  • Why is only the post-operative state investigated? From a clinical point of view, the pre-operative sample would be more convenient. In the post-operative sample, the non-tumor cell-free DNA could potentially increase, “diluting” the ctDNA and decreasing the detection rate. Certainly, preoperative selection of patients prone to recurrence will allow physicians to tailor therapies, especially in the context of neoadjuvant chemotherapy.  However, several studies have been conducted, but failed to find an association between preoperative ctDNA and postoperative outcomes, particularly in non-metastatic CRC (Schraa SJ, van Rooijen KL, Koopman M, Vink GR, Fijneman RJA. 2001. Cell-free circulating (tumor) DNA before surgery as a prognostic factor in non-metastatic colorectal cancer: A systematic review. Cancer (Basel). 2022 Apr 29;14(9):2218. doi: 10.3390/cancers14092218. PMID: 35565347; PMCID: PMC9101623; Riethdorf S, O'Flaherty L, Hille C, Pantel K. Clinical applications of the CellSearch platform in cancer patients. Adv Drug Deliv Rev. 2018 Feb 1;125:102-121. doi: 10.1016/j.addr.2018.01.011. Epub 2018 Feb 2. PMID: 29355669). Therefore, to specifically study stage II CRC, we decided to focus on the few existing studies that evaluated ctDNA in a postoperative setting, leaving the preoperative setting for further evidence confirmation. Finally, post-operative ctDNA measurement was better in a clinical setting for different reasons: - Improved ability to detect minimal residual disease (If ctDNA is detectable after surgery, it means that the tumor has not been completely eradicated and the patient is at high risk of recurrence) - Greater prognostic value than preoperative ctDNA (Preoperative ctDNA is often present in patients with advanced cancer, but does not provide information on the success of surgical resection) - Clinical implications, Clinical implications for adjuvant therapy (preoperative ctDNA is not useful for making postoperative treatment decisions because it does not distinguish between completely resected and residual disease. ) - Reduced risk of confounding factors (preoperative ctDNA can be influenced by factors such as tumor volume, tumor necrosis, and inflammatory response, making it difficult to interpret prognostically). I added an explanation in the discussion section Line 327-333.
  • Carefully check all aberrations are spelled out only the first time they are used (for instance CEA on line 381), and listed below each figure and table (for instance “f.up” in Table 1). We have now carefully checked the manuscript to ensure that all abbreviations are spelled out only the first time they appear in the text and are consistently used thereafter. Additionally, we have reviewed all figures and tables, ensuring that abbreviations are appropriately listed below each figure and table for clarity. 
  • I find the layout of Table 1 hard to follow. Should it be two tables, or can the information be fused into one table? To improve clarity and ease of interpretation, we have reformatted the table into a single horizontal layout, ensuring that all key data are presented in a more structured and accessible manner. This approach maintains the cohesion of information while enhancing readability. Additionally, we have: Adjusted column alignment for better data visualization, Revised text spacing and abbreviations to improve clarity, Enhanced the table legend to provide clearer guidance on data interpretation.
  • Are there other studies, for other carcinomas, that this study could be compared to? Are there any studies suggesting the post-operative sample might not be associated to prognosis? Any pre-operative investigations from stage II CRC? Most studies indicate that the presence of circulating tumor DNA (ctDNA) after surgery is a strong prognostic indicator of recurrence in patients with stage II colon cancer. However, some studies have investigated the use of ctDNA as a predictive biomarker to guide treatment decisions with conflicting results. For example, the COBRA study (NCT04068103) evaluated the efficacy of ctDNA in guiding the administration of adjuvant chemotherapy in patients with low-risk stage II colon cancer. The results showed that the use of ctDNA as a predictive biomarker did not significantly improve the benefit of adjuvant chemotherapy in these patients. However, a tumor agnostic methodology was used in this study and the authors acknowledge that an important bias resulting from this contributed to the discontinuation of the study (Morris VK, Yothers G, Kopetz S, Puhalla SL, Lucas PC, Iqbal A, et al. Phase II results of circulating tumor DNA as a predictive biomarker in adjuvant chemotherapy in patients with stage II colon cancer: The NRG-GI005 (COBRA) phase II/III trial. Journal of Clinical Oncology. 2024 Jan 20;42(3_suppl):5-5. DOI: 10.1200/JCO.2024.42.3_suppl.5). There is low evidence that preoperative ctDNA measurement is associated with the risk of recurrence (Benhaim L, Bouché O, Normand C, Didelot A, Mulot C, Le Corre D, Garrigou S, Djadi-Prat J, Wang-Renault SF, Perez-Toralla K, Pekin D, Poulet G, Landi B, Taieb J, Selvy M, Emile JF, Lecomte T, Blons H, Chatellier G, Link DR, Taly V, Laurent-Puig P, et al. Circulating tumor DNA is a prognostic marker for tumor recurrence in stage II and III colorectal cancer: a multicenter prospective cohort study (ALGECOLS). Eur J Cancer. 2021 Dec;159:24-33. doi: 10.1016/j.ejca.2021.09.004. Epub 2021 Oct 30. PMID: 34731746.). However, the use of postoperative ctDNA is preferable to preoperative ctDNA in the study of colorectal cancer for several important reasons: - Improved ability to detect minimal residual disease (If ctDNA is detectable after surgery, it means that the tumor has not been completely eradicated and the patient is at high risk of recurrence) - Greater prognostic value than preoperative ctDNA (Preoperative ctDNA is often present in patients with advanced cancer, but does not provide information on the success of surgical resection) - Clinical implications, Clinical implications for adjuvant therapy (preoperative ctDNA is not useful for making postoperative treatment decisions because it does not distinguish between completely resected and residual disease. ) - Reduced risk of confounding factors (preoperative ctDNA can be influenced by factors such as tumor volume, tumor necrosis, and inflammatory response, making it difficult to interpret prognostically).
  • The number of patients with positive post-operative ctDNA are few. The usefulness of this test thereby decreases, since most patients will not have a positive sample, and within that “negative” group there will still be patients with a high risk for relapse. This is not covered in the Discussion. We acknowledge that the proportion of ctDNA-positive patients in the included studies is relatively low, which may affect the overall predictive value of this biomarker. We agree that within the ctDNA-negative group, there may still be patients at high risk for relapse due to other biological or clinicopathological factors. This limitation underscores the need for further refinement of risk stratification approaches, potentially integrating ctDNA with additional biomarkers or clinical parameters. To address this concern, we have now expanded the Discussion section to better highlight the challenges related to false-negative cases and the necessity of complementary prognostic tools (Line 446-461). 
  • In the tumor-informed approaches, which targets did they find? Any recurrent hotspot variants? Recurrent genes? In our review, the majority of studies employing tumor-informed ctDNA detection focused on identifying patient-specific somatic mutations derived from tumor tissue sequencing. The most commonly targeted recurrent genes and hotspot variants included: TP53, KRAS, and NRAS mutations, which are frequently altered in colorectal cancer (CRC); APC and PIK3CA mutations, which are known drivers of CRC progression; BRAF V600E, associated with poor prognosis in CRC. These tumor-informed approaches used techniques such as next-generation sequencing (NGS) and droplet digital PCR (ddPCR) to track individualized variants in post-surgical plasma samples. To better address your suggestion, we have now expanded our Discussion section to include a more detailed summary of the recurrent genes and mutations that were reported in the analyzed studies. (Line 374-387).
  • Minor comment: Line 126-127, please check tense. Regarding the minor comment on Line 126-127, we have reviewed and corrected the tense to ensure grammatical accuracy and consistency.

Reviewer 2 Report

Comments and Suggestions for Authors

Abstract is good

Supplementary data

Should add the doi of the mentioned publication

Introduction:

Abbreviation list should be added

More information about ctDNA and CRC should be added

I was good but a clear explanation of the aim should be added to explain it was a review or publication paper

Figures

More figures about the statistical studies should be added

References are good

Author Response

Dear Reviewer 2,

I sincerely appreciate the time and effort you have dedicated to reviewing our manuscript and for your valuable suggestions. We have carefully considered all your comments and have addressed each point accordingly, making the necessary revisions to improve the quality of our work.

We hope that the changes meet your expectations, and we remain available for any further clarifications.

Red (Reviewer) Green (Author)

Abstract is good

Supplementary data

Should add the doi of the mentioned publication. I have added the doi to the reference list.

Introduction:

Abbreviation list should be added. I have added “Abbreviation List” at the end of the introduction.

More information about ctDNA and CRC should be added. I have added paragraphs in the Introduction section to better explain the role of ctDNA in CRC risk stratification (highlighted in green in the text). Line 58-65; 67-71; 76-81.   

Colorectal cancer (CRC) is one of the most prevalent and deadly malignancies globally, accounting for 1.9 million new cases and 935,000 deaths annually (1). Although surgery remains the primary curative treatment for resectable CRC, recurrence poses a significant challenge, affecting approximately 30-50% of patients. Notably, nearly 25 percent of recurrences occur in individuals initially diagnosed with stage II disease, which represents around 30% of all CRC cases (2,3). The 8th edition of the TNM classification system stratifies stage II patients into low-risk and high-risk categories based on clinicopathological factors such as lymphovascular invasion, poor tumor differentiation, and T4 tumor stage. High-risk patients are considered candidates for adjuvant chemotherapy (adj-CT) (4). However, there is a significant lack of clarity regarding which patients will truly benefit from postoperative therapies. Many patients with high-risk features do not experience recurrence, while some considered low-risk unexpectedly relapse. This uncertainty leads to suboptimal treatment decisions, with a substantial proportion of patients unnecessarily exposed to the potential toxicities of adj-CT (5–8). Therefore, these issues highlight the urgent need for a robust and more effective method to predict treatment response and facilitate relapse detection. Minimal residual disease (MRD) is defined as the presence of a small number of neoplastic cells in the bloodstream after curative treatment. While these residual cells are not detectable using standard diagnostic techniques, they can be indirectly identified using highly sensitive methods such as circulating tumor DNA (ctDNA) analysis (9). ctDNA, which consists of short DNA fragments shed by apoptotic and necrotic tumor cells into the bloodstream, serves as a real-time molecular biomarker reflecting the presence of residual disease and tumor dynamics. Unlike traditional biomarkers, ctDNA allows for a non-invasive assessment of tumor burden, clonal evolution, and treatment response, making it an attractive tool for precision oncology. Recent advancements in sequencing technologies, such as digital droplet PCR (ddPCR) and next-generation sequencing (NGS), have significantly enhanced the sensitivity and specificity of ctDNA detection. By analyzing tumor-derived genetic material from peripheral blood, ctDNA can provide molecular evidence of recurrence months before its clinical or radiological confirmation (10,11). Several studies have demonstrated that ctDNA detection post-surgery is associated with a markedly higher risk of relapse, even in patients considered to be at low clinical risk. Moreover, ctDNA can capture minimal tumor burden that conventional imaging modalities fail to detect, reinforcing its role as a powerful prognostic indicator. Observational studies in patients with solid tumors have shown a very high risk of recurrence when ctDNA is detected after curative-intent surgery (12–14). However, its role in guiding treatment decisions in CRC remains controversial. Most existing studies have focused on metastatic disease and often fail to differentiate between CRC stages, particularly in the perioperative setting (15,16). Emerging evidence suggests that ctDNA can be utilized for dynamic risk stratification, allowing for tailored treatment approaches based on real-time molecular assessment rather than solely on clinicopathological features(17). This individualized approach has the potential to refine adjuvant therapy recommendations, preventing overtreatment in patients who may not benefit from additional chemotherapy while ensuring that high-risk patients receive timely intervention. To address this knowledge gap, the present study conducts a systematic review and meta-analysis of recent research on the prognostic value of postoperative ctDNA in stage II CRC.Specifically, we will explore three main topics: a) the association between postoperative ctDNA positivity and recurrence risk in patients who have undergone curative-intent surgery for CRC; b) the prognostic value of ctDNA positivity following adj-CT; c) the utility of ctDNA as a dynamic tool for monitor recurrence during follow-up.  This review will provide a comprehensive analysis of existing literature to clarify the clinical relevance of ctDNA as a biomarker for recurrence risk assessment and treatment stratification in stage II CRC. By addressing current knowledge gaps, this study aims to contribute to more precise and personalized therapeutic strategies for CRC management.

It was good but a clear explanation of the aim should be added to explain it was a review or publication paper. I have added this sentence at the end of the introduction to better clarify our aims and scope: "This review will provide a comprehensive analysis of the existing literature to clarify the clinical relevance of ctDNA as a biomarker for recurrence risk assessment and treatment stratification in stage II CRC. By addressing current knowledge gaps, this study aims to contribute to more precise and personalized therapeutic strategies for CRC management". Line 87-91

Figures

More figures about the statistical studies should be added. We acknowledge that although a forest plot provides a comprehensive summary of the meta-analysis results, additional graphical representations such as funnel plots and sensitivity analyses can further enhance the clarity of statistical interpretations. However, we recognize that a funnel plot may not be appropriate in cases where fewer than ten studies are included in the meta-analysis, as its reliability in detecting publication bias is significantly reduced with small sample sizes. In addition, sensitivity analysis may not be feasible in cases where all included studies show homogeneous results or where the dataset lacks sufficient variability to perform meaningful sensitivity tests. In addition, we chose to include subgroup analyses to explore potential sources of heterogeneity and to better understand how different patient characteristics, treatment regimens, and ctDNA analysis methods might affect the results.

References are good